# Inhibition of Transient Receptor Potential Melastatin 3 ion channels by G-protein βγ subunits

**Doreen Badheka[1†], Yevgen Yudin[1†], Istvan Borbiro[1], Cassandra M Hartle[2], Aysenur Yazici[1], Tooraj Mirshahi[2], Tibor Rohacs[1]***

[1]New Jersey Medical School, Rutgers, the State University of New Jersey, Newark, United States; [2]Department of Molecular and Functional Genomics, Weis Center for Research, Geisinger Clinic, Danville, United States

**Abstract** Transient receptor potential melastatin 3 (TRPM3) channels are activated by heat, and chemical ligands such as pregnenolone sulphate (PregS) and CIM0216. Here, we show that activation of receptors coupled to heterotrimeric Gi/o proteins inhibits TRPM3 channels. This inhibition was alleviated by co-expression of proteins that bind the βγ subunits of heterotrimeric G-proteins (Gβγ). Co-expression of Gβγ, but not constitutively active Gαi or Gαo, inhibited TRPM3 currents. TRPM3 co-immunoprecipitated with Gβ, and purified Gβγ proteins applied to excised inside-out patches inhibited TRPM3 currents, indicating a direct effect. Baclofen and somatostatin, agonists of Gi-coupled receptors, inhibited $Ca^{2+}$ signals induced by PregS and CIM0216 in mouse dorsal root ganglion (DRG) neurons. The $GABA_B$ receptor agonist baclofen also inhibited inward currents induced by CIM0216 in DRG neurons, and nocifensive responses elicited by this TRPM3 agonist in mice. Our data uncover a novel signaling mechanism regulating TRPM3 channels.

*For correspondence: rohacsti@njms.rutgers.edu

†These authors contributed equally to this work

Competing interests: The authors declare that no competing interests exist.

## Introduction

Transient receptor potential melastatin 3 (TRPM3) channels are activated by heat (*Vriens et al., 2011*), and a number of chemical ligands such as pregnenolone sulphate (PregS) (*Oberwinkler and Philipp, 2014*) and the newly described synthetic agonist CIM0216 (*Held et al., 2015*). These channels were shown to act as heat sensors in dorsal root ganglion (DRG) neurons; mice lacking TRPM3 had altered behavioral responses to noxious heat (*Vriens et al., 2011*). TRPM3 is also expressed in a variety of other tissues, including the brain, kidneys and pancreatic β-cells (*Oberwinkler and Philipp, 2014*).

The βγ subunits of heterotrimeric G-proteins were originally thought to be scaffolds for the Gα subunits, keeping them inactive in non-stimulated cells. Seminal work on cardiac G-protein activated $K^+$ (GIRK) channels demonstrated important direct physiological roles for Gβγ (*Logothetis et al., 1987*). All GIRK channels (Kir3.1–3.4) are activated by cell surface receptors that couple to heterotrimeric Gi/o proteins, via direct binding of Gβγ to the channel. This effect plays roles in slowing the heart rate by muscarinic stimulation, and in the analgesic effects of opioids (*Hibino et al., 2010*).

We and others have shown recently that in various cellular expression systems PregS-induced TRPM3 activity requires the presence of the membrane phospholipid phosphatidylinositol 4,5-bisphosphate [PI(4,5)P$_2$] (*Badheka et al., 2015*; *Tóth et al., 2015*), which is a common feature of most TRP channels (*Rohacs, 2014*). Stimulation of plasma membrane receptors that induce PI(4,5)P$_2$ hydrolysis via phospholipase C (PLC) activation, was shown to inhibit both heterologously expressed TRPM3 channels (*Badheka et al., 2015*; *Tóth et al., 2015*) and endogenous TRPM3 in insulinoma cells (*Tóth et al., 2015*). The purified TRPM3 protein in planar lipid bilayers also required PI(4,5)P$_2$

for activity induced by PregS (*Uchida et al., 2016*). Other activators of the channel, nifedipine, or the combination of PregS plus clotrimazole, however, could induce activity in bilayers in the absence of PI(4,5)P$_2$, showing that the requirement for this lipid may be modality dependent (*Uchida et al., 2016*).

Here, we report that dialysis of PI(4,5)P$_2$ via the patch pipette did not alleviate inhibition of PregS-induced TRPM3 currents by Gq-coupled receptors, but co-expression of a protein that binds G$\beta\gamma$, thus acts as a 'G$\beta\gamma$ sink', significantly attenuated it. TRPM3 currents were also robustly inhibited by activation of various Gi-coupled receptors, and that effect was also attenuated by G$\beta\gamma$ sinks. Coexpression of G$\beta\gamma$, but not G$\alpha$i or G$\alpha$o, in intact cells inhibited TRPM3 currents. TRPM3 co-immunoprecipitated with G$\beta$, and application of purified G$\beta\gamma$ to excised inside-out patches inhibited the channel, indicating a direct effect. PregS-induced Ca$^{2+}$ signals in DRG neurons were inhibited by stimulating Gi-coupled receptors with somatostatin or by the GABA$_B$ receptor agonist baclofen. Baclofen also inhibited Ca$^{2+}$ signals and currents evoked by CIM0216, as well as nocifensive behavioral responses induced by this TRPM3 agonist in vivo. Our data identify TRPM3 as a novel ion channel target of G$\beta\gamma$ subunits.

## Results

### Inhibition of TRPM3 by Gq- and Gi-coupled receptors via G$\beta\gamma$

It was recently shown by two different laboratories that TRPM3 channels require PI(4,5)P$_2$ for activity, and that depletion of phosphoinositides using various inducible phosphatases inhibited PregS-induced TRPM3 currents (*Badheka et al., 2015*; *Tóth et al., 2015*). Accordingly, we also found that stimulating M1 muscarinic receptors that couple to PLC inhibited TRPM3; interestingly, this inhibition was faster and more robust than inhibition by the inducible phosphoinositide phosphatases (*Badheka et al., 2015*).

To assess the involvement of PI(4,5)P$_2$ depletion in PLC mediated inhibition, here we performed whole-cell patch clamp measurements in HEK293 cells cotransfected with the Gq-coupled M1 muscarinic receptors and TRPM3. We supplemented the patch pipette with PI(4,5)P$_2$, and compared carbachol-induced inhibition to control cells without the lipid in the patch pipette. We used this maneuver earlier to alleviate inhibition of the activity of several different ion channels by PI(4,5)P$_2$ depletion (*Borbiro et al., 2015*; *Lukacs et al., 2007*, *2013*). *Figure 1A* shows a whole-cell patch clamp measurement, using a ramp protocol from −100 to 100 mV; current values at −100 and +100 mV are plotted. PregS (50 µM) induced an outwardly rectifying current that decreased over time and stabilized at a quasi steady state. Application of 100 µM carbachol induced a fast and complete inhibition. In cells dialyzed with 100 µM diC$_8$ PI(4,5)P$_2$ carbachol-induced inhibition was somewhat smaller, but this difference was not statistically significant (p=0.103) (*Figure 1B,C*). We obtained similar results using bradykinin, where inclusion of PI(4,5)P$_2$ did not significantly alleviate inhibition by this compound in cells expressing TRPM3 and the B2 bradykinin receptor (data not shown).

These data indicate that pathways other than PI(4,5)P$_2$ depletion play important roles in inhibition of TRPM3 currents by PLC-coupled receptors. G-protein-coupled receptors (GPCRs) activate PLC$\beta$ isoforms via heterotrimeric G-proteins in the Gq/11 family. To test the possible involvement of G-protein subunits, we co-expressed the C-terminal domain of the $\beta$-adrenergic receptor kinase ($\beta$ARK-CT), which binds G$\beta\gamma$ subunits and has been used earlier to 'sink' G$\beta\gamma$ and thus alleviate effects mediated by this subunit (*He et al., 1999*; *Yamauchi et al., 2000*). *Figure 1D–F* shows that co-expressing the $\beta$ARK-CT construct significantly attenuated the inhibitory effect of M1 receptor activation by 5 µM Acetylcholine (ACh).

G$\beta\gamma$ subunits are not specific to Gq-coupled receptors, indeed most G$\beta\gamma$-mediated biological effects, such as GIRK channel activation, are initiated by activation of receptors that act via the Gi/o family. Thus, we co-expressed TRPM3 and the Gi-coupled M2 muscarinic receptors in HEK293 cells, and tested the effect of activating those receptors. *Figure 1G* shows that ACh quickly and completely inhibited PregS-induced TRPM3 currents in cells expressing M2 receptors. Next, we tested if Gi-mediated inhibition involves G$\beta\gamma$. *Figure 1H,I* shows that co-expression of $\beta$ARK-CT significantly attenuated ACh-mediated inhibition. The inhibitory effect of ACh was also alleviated by a different G$\beta\gamma$ sink, the inactivated G203A mutant of the G$\alpha$i3 protein (*Ogier-Denis et al., 1996*) (*Figure 1I*).

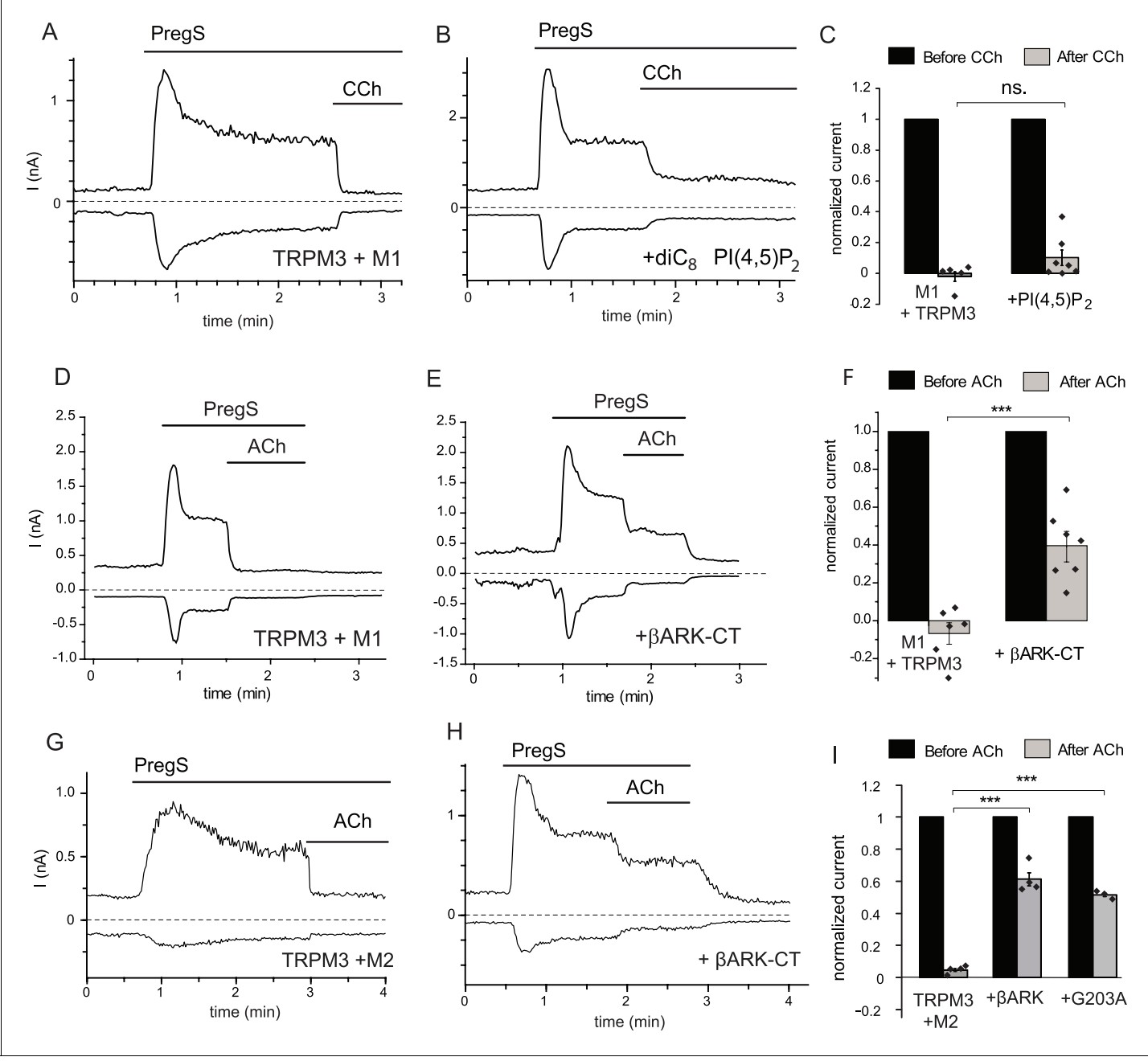

**Figure 1.** Inhibition of TRPM3 by Gq-coupled M1 and Gi-coupled M2 muscarinic receptors via G$\beta\gamma$. Whole-cell patch clamp experiments on HEK cells expressing mTRPM3$\alpha$2 and Gq-coupled M1 or Gi-coupled M2 muscarinic receptors were performed as described in Materials and methods. TRPM3 currents were evoked by 50 µM PregS, currents are plotted at −100 and 100 mV (lower and upper traces), dashed lines show zero current. (**A–B**) Representative traces for inhibition by 100 µM carbachol (CCh), without (**A**) or with 100 µM diC$_8$ PI(4,5)P$_2$ (**B**) in the whole-cell patch pipette in cells expressing M1 muscarinic receptors. (**C**) Summary of the data (n = 5 for control and n = 7 for PI(4,5)P$_2$, ns: p=0.103, two sample t-test). (**D**) Representative trace showing inhibition by 5 µM ACh, in a cell expressing M1 muscarinic receptors (**E**) similar experiment in a cell co-expressing the C-terminus of $\beta$ARK which binds to G$\beta\gamma$. (**F**) Summary data (n = 6 for control and n = 7 for $\beta$ARK-CT, ***p=0.00032, two sample t-test). (**G**) Representative trace showing inhibition by 5 µM ACh in a cell expressing the Gi-coupled M2 muscarinic receptors and mTRPM3$\alpha$2, (**H**) similar experiment in a cell co-expressing the C-terminus of $\beta$ARK. (**I**) Summary data, (n = 4 for control, n = 4 for $\beta$ARK-CT, n = 3 for G203A). ***p=0.000003 and p=0.000022, one-way analysis of variance with Bonferroni post hoc comparison.

The following figure supplements are available for figure 1:

**Figure supplement 1.** Activation of M1, but not M2 muscarinic receptors induces PI(4,5)P$_2$ hydrolysis.

*Figure 1 continued on next page*

*Figure 1 continued*

**Figure supplement 2.** Activation of GPCRs inhibit TRPM3 currents in various conditions.

**Figure supplement 3.** PLCγ activation by the PDGFRβ inhibits TRPM3 activity.

Next, we tested if overexpressed Gi-coupled M2 receptors induce any detectable PLC activation. We transfected HEK293 with M1, or M2 receptors, and a pair of fluorescence resonance energy transfer (FRET)-based $PI(4,5)P_2$ sensors, the CFP- and YFP-tagged tubby domain (*Borbiro et al., 2015*; *Quinn et al., 2008*). *Figure 1—figure supplement 1* shows that application of carbachol induced a significant decrease in FRET in cells transfected with M1 receptors, indicating a decrease in $PI(4,5)P_2$ levels, whereas in cells transfected with M2 receptors, $PI(4,5)P_2$ levels did not change. These data show that overexpressed M2 receptors do not signal to PLC and that endogenous Gq-coupled muscarinic receptors in HEK cells do not express at sufficiently high levels to induce a significant decrease in $PI(4,5)P_2$ levels. These results show that PLC activation is not necessary for inhibition of TRPM3 upon GPCR activation.

The inhibitory effect of muscarinic M1 or M2 receptor activation on TRPM3 did not depend on the presence of extracellular $Ca^{2+}$, as ACh and carbachol inhibited PregS-induced TRPM3 currents in the absence of extracellular $Ca^{2+}$ (*Figure 1—figure supplement 2A,B*). TRPM3 channels have an alternative permeation pathway that is open when clotrimazole and PregS are co-applied (*Vriens et al., 2014*). This alternative pathway displays lower level of inward rectification, and thus higher current levels at negative voltages. *Figure 1—figure supplement 2C* shows that currents induced by clotrimazole/PregS were also fully inhibited by ACh. We also tested if activation of the Gi-coupled D2 Dopamine receptors inhibited TRPM3 currents. *Figure 1—figure supplement 2D* shows that application of quinpirole to cells expressing D2 and TRPM3 resulted in complete inhibition of TRPM3 currents induced by either PregS, or the combination of PregS and clotrimazole. Overall, these data show that activation of the Gi-coupled M2 muscarinic, or D2 dopamine receptors inhibit TRPM3 currents under a variety of experimental conditions and channel activation modalities.

Our data so far suggest that G-protein βγ subunits play an important role in TRPM3 current inhibition upon M1 muscarinic receptor activation. We found no clear evidence for the role of $PI(4,5)P_2$ hydrolysis, potentially due to the masking effect of the robust inhibition by Gβγ. To test the effect of PLC activation on TRPM3 currents without the release of Gβγ subunits, we co-expressed TRPM3 with the receptor tyrosine kinase platelet-derived growth factor (PDGF) β receptor (PDGFRβ), which couples to PLCγ. As a negative control, we co-expressed TRPM3 with the Y1009F-Y1021F mutant of PDGFRβ that does not activate PLC (*Ridefelt and Siegbahn, 1998*; *Rohács et al., 2005*). *Figure 1—figure supplement 3* shows that PDGF inhibited PregS-induced currents in Xenopus oocytes co-expressing TRPM3 and PDGFRβ, but not in cells expressing the Y1009F Y1021F mutant. These data show that in principle, PLC activation is sufficient to inhibit TRPM3 activity in the absence of G-protein activation. For the rest of this study, we focus on Gi-coupled receptor activation to avoid confounding effects of PLC activation.

## Inhibition of TRPM3 by Gβγ but not Gαi or Gαo subunits

Our data so far indicate the involvement of Gβγ subunits in inhibiting TRPM3 channels. To assess their role more directly, we co-expressed Gβ1 and Gγ2 with TRPM3 in *Xenopus laevis* oocytes, and compared them to control oocytes injected with RNA encoding TRPM3. Co-expression of Gβ1γ2 significantly inhibited TRPM3 currents (*Figure 2A–C*). To test the potential role of Gα subunits, we also coexpressed the wild type Gαi3, and the constitutively active G205L mutant of Gαi2 and the same G205L mutant of Gαo (*Hermouet et al., 1991*). Neither the wild type nor the constitutively active mutant Gα subunits inhibited PregS-induced TRPM3 activity (*Figure 2D*). These data indicate that Gβγ, but not Gα subunits inhibit TRPM3 channels. We also tested the effect of Gβ5, a subunit, which does not potentiate GIRK channels (*Mirshahi et al., 2002*), and found that it had no inhibitory effect on TRPM3 when co-expressed with Gγ2 (*Figure 2D*).

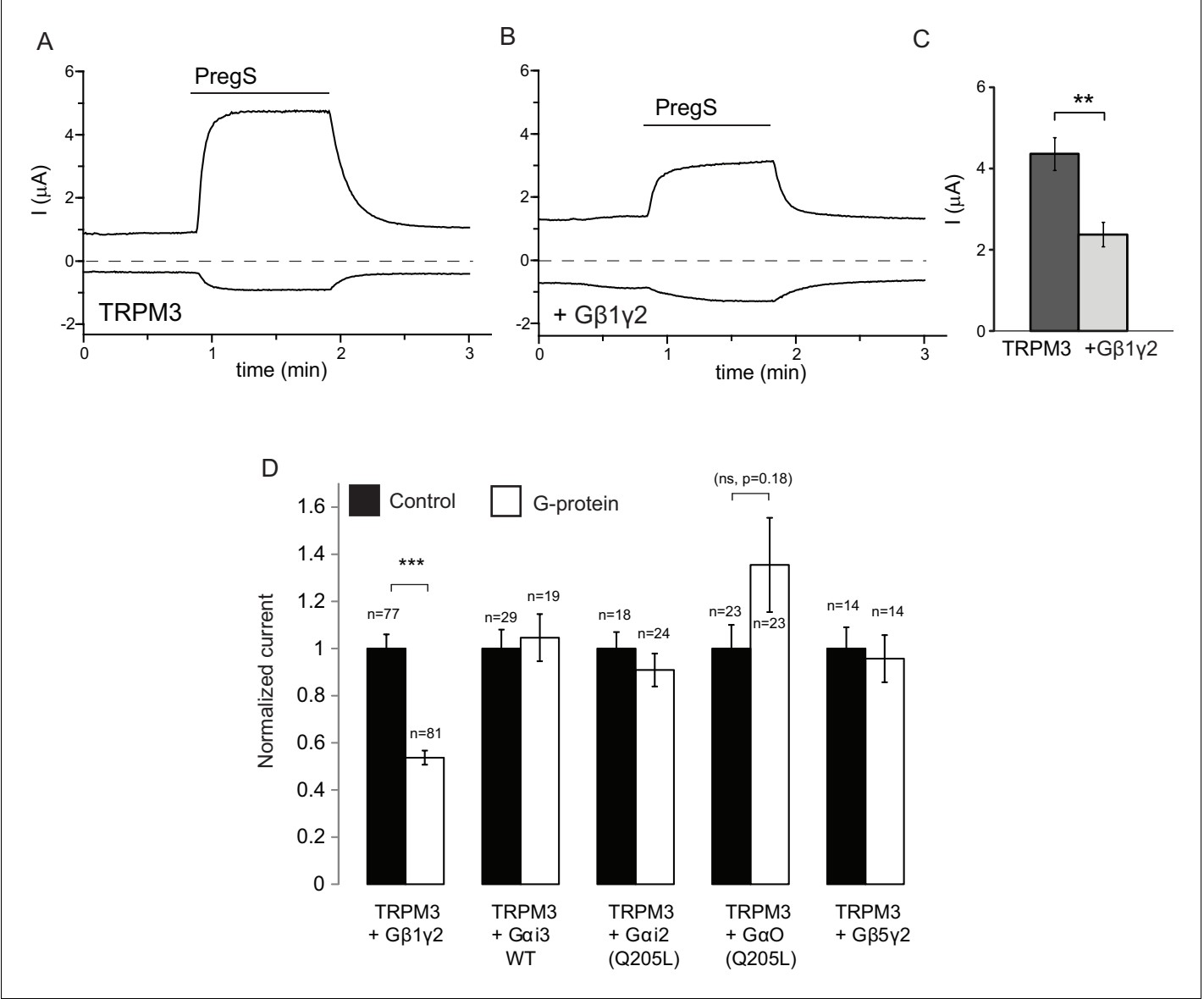

**Figure 2.** Co-expressed Gβ1γ2, but not Gαi or Gαo inhibits TRPM3 currents. TEVC measurements in Xenopus oocytes expressing hTRPM3 were performed as described in Materials and methods; currents are plotted at 100 mV (upper traces) and −100 mV (lower trace). Currents were evoked by 50 µM PregS in control oocytes (**A**) and in oocytes expressing Gβ1γ2 (**B**). (**C**) Summary data for current amplitudes at 100 mV (n = 17 for each groups from one representative experimental day) (**D**) Normalized PregS-induced current amplitudes in oocytes co-expressing hTRPM3 and different G-protein constructs at 100 mV. Black bars are normalized current levels for control hTRPM3 expressing oocytes (see Materials and methods for details), empty bars are normalized current levels for oocytes also expressing the various G-protein subunits. The number of measurements on individual oocytes are indicated for each group. Statistical analysis was performed with two sample t-test ***p<0.005, corrected for multiple comparisons.

The following figure supplement is available for figure 2:

**Figure supplement 1.** Co-expressed Gβ1γ2, but not Gαi or Gαo inhibits hTRPM3 currents; box and scatter plots.

Next, we tested the effects of purified Gβγ subunits directly applied to excised inside-out patches. Consistent with earlier results (*Badheka et al., 2015*), TRPM3 currents displayed a substantial rundown in excised patches, after a transient initial increase upon patch excision (*Figure 3A,B*). We showed earlier that this current rundown is caused by the decrease of endogenous PI(4,5)P$_2$ levels in the patch membrane (*Badheka et al., 2015*). Accordingly, channel activity was restored by the

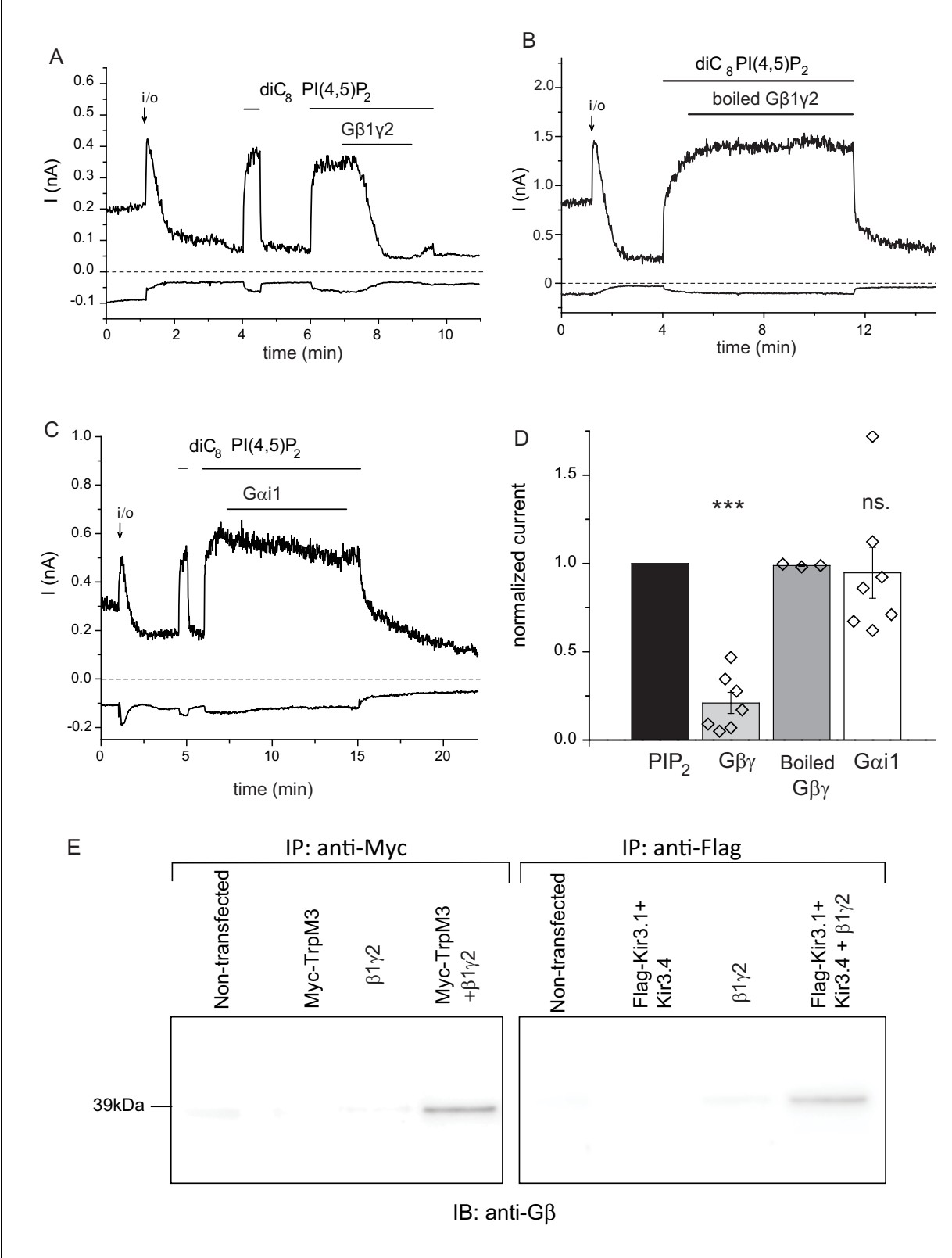

**Figure 3.** Purified recombinant Gβ1γ2 inhibits TRPM3 currents in excised patches. (**A–C**) Excised inside-out patch clamp experiments were performed in Xenopus oocytes expressing hTRPM3, with 100 μM PregS in the patch pipette, as described in Materials and methods, currents at −100 mV (lower traces) and 100 mV (upper traces) are shown. The establishment of the inside-out (i/o) configuration is marked with the arrow, the application of 25 μM diC$_8$ PI(4,5)P$_2$ is shown with the horizontal line. (**A**) the effect of intact Gβ1γ2 (50 ng/ml), (**B**) the effect of Gβ1γ2 boiled for 15 min before the experiment. *Figure 3 continued on next page*

*Figure 3 continued*

(C) The effect of 50 ng/ml Gαi1 (D) Summary of the data, the effects of the G-proteins were normalized to the currents induced by PI(4,5)P$_2$ before the application the G-protein (n = 3 for boiled Gβγ, n = 7 for Gβγ and for Gαi1). (E) Co-immunoprecipitation of myc-TRPM3 (left panel) and flag-Kir3.4 was performed as described in the materials and methods section. HEK cells were transfected with the constructs indicated, immunoprecipitated using an anti-myc (left) or anti-flag antibody, and immunoblotted with an anti-Gβ antibody. Blots are representatives for four independent experiments, from four different transfections. Statistical analysis for the electrophysiological experiments was performed with one sample t-test ***p<0.00001, ns: p=0.72.

The following figure supplement is available for figure 3:

**Figure supplement 1.** Inhibition TRPM3 in excised patches by Gβγ purified from bovine brain.

application of diC$_8$ PI(4,5)P$_2$, and when purified recombinant Gβ1γ2 (50 ng/ml) was applied to the patch in the continued presence of PI(4,5)P$_2$, currents were inhibited (*Figure 3A,D*). The inhibition developed slowly, but it was almost complete in most patches. Boiled Gβγ applied in the same protocol had no inhibitory effect (*Figure 3B,D*), and purified Gαi1 did not inhibit channel activity either (*Figure 3C,D*). We also tested the effect of a different Gβγ preparation purified from bovine brain, which had a similar, although faster developing inhibitory effect on TRPM3 currents in excised patches (*Figure 3—figure supplement 1*).

To demonstrate direct interaction between Gβγ and TRPM3, we co-immunoprecipitated the two proteins (*Figure 3E*). When HEK cells were co-transfected with the myc-tagged TRPM3 and Gβ1γ2, we could detect Gβ using an anti-Gβ antibody in anti-myc immunoprecipitates. Gβ was not detected after immunoprecipitation with the anti-myc antibody from non-transfected cells, from cells transfected with Gβ1γ2, or cells transfected with myc-TRPM3 only (*Figure 3E*, left panel). In control experiments, we also co-immunoprecipitated Gβγ with the flag-tagged Kir3.4 (GIRK4) the well-established Gβγ regulated ion channel. Similarly to the behavior of TRPM3, Gβ was only detected in anti-flag immunoprecipitates, when Gβ1γ2, and the flag-tagged Kir3.4 were co-transfected (*Figure 3E*, right panel). A likely explanation for these data is that endogenous Gβγ binds preferentially to Gα, and the interaction can only be detected when excess Gβγ is present.

## Inhibition of TRPM3 activity in DRG neurons by Gi-coupled receptors

TRPM3 channels are found mainly in small nociceptive DRG neurons. These neurons express a number of different Gi/o coupled receptors, including opioid receptors, somatostatin receptors, neuropeptide Y and GABA$_B$ receptors. The highest expressing of these at the RNA level are GABA$_B$ receptors (both type 1 and 2) (*Thakur et al., 2014*); somatostatin (SST) receptors type 1 and 2 are expressed at lower levels (*Thakur et al., 2014*). Both GABA$_B$ (*Hanack et al., 2015*), and SST (*Pintér et al., 2006*) receptor activation has been implicated in regulating pain, thus we focused on these two receptor types. DRG neurons are highly heterogeneous, but to our knowledge no TRPM3 reporter mouse is available to identify cells expressing these channels. TRPM3 RNA shows substantial enrichment in a subpopulation of small peptidergic TRPM8 positive neurons (PEP1) (*Usoskin et al., 2015*). Here, we used a transgenic mouse line in which the promoter of TRPM8 drives GFP expression (*Takashima et al., 2007*; *Yudin et al., 2016*), to assess if this reporter mouse is useful in identifying TRPM3 positive DRG neurons.

*Figure 4A–D* shows that repetitive short (60 s) applications of PregS (12.5 μM) evoked Ca$^{2+}$ signals in many DRG neurons. *Figure 4—figure supplement 1* shows the responsiveness of GFP-negative and GFP-positive neurons. About 20% of GFP-negative neurons responded to 12.5 μM PregS. The responsiveness of GFP-positive neurons was higher, ~75% of smaller (diameter <22.5 μm) and ~45% of larger (>22.5 μm) cells responded to 12.5 μM PregS. We found earlier that most small GFP-positive neurons responded not only to TRPM8 agonists, but also to capsaicin, a TRPV1 agonist (*Yudin et al., 2016*), thus small GFP positive neurons likely correspond to PEP1 neurons, which express TRPM8, TRPM3 and TRPV1 (*Usoskin et al., 2015*). Application of 1 μM somatostatin inhibited PregS-induced Ca$^{2+}$ signals in a subpopulation of DRG neurons (27 out of 65 cells, 41.5%) (*Figure 4B*). *Figure 4—figure supplement 2* shows representative images as well as representative traces for individual cells. We also tested neuropeptide Y in a small number of cells, this peptide inhibited PregS-induced Ca$^{2+}$ signals in 4 out of 9 neurons (data not shown).

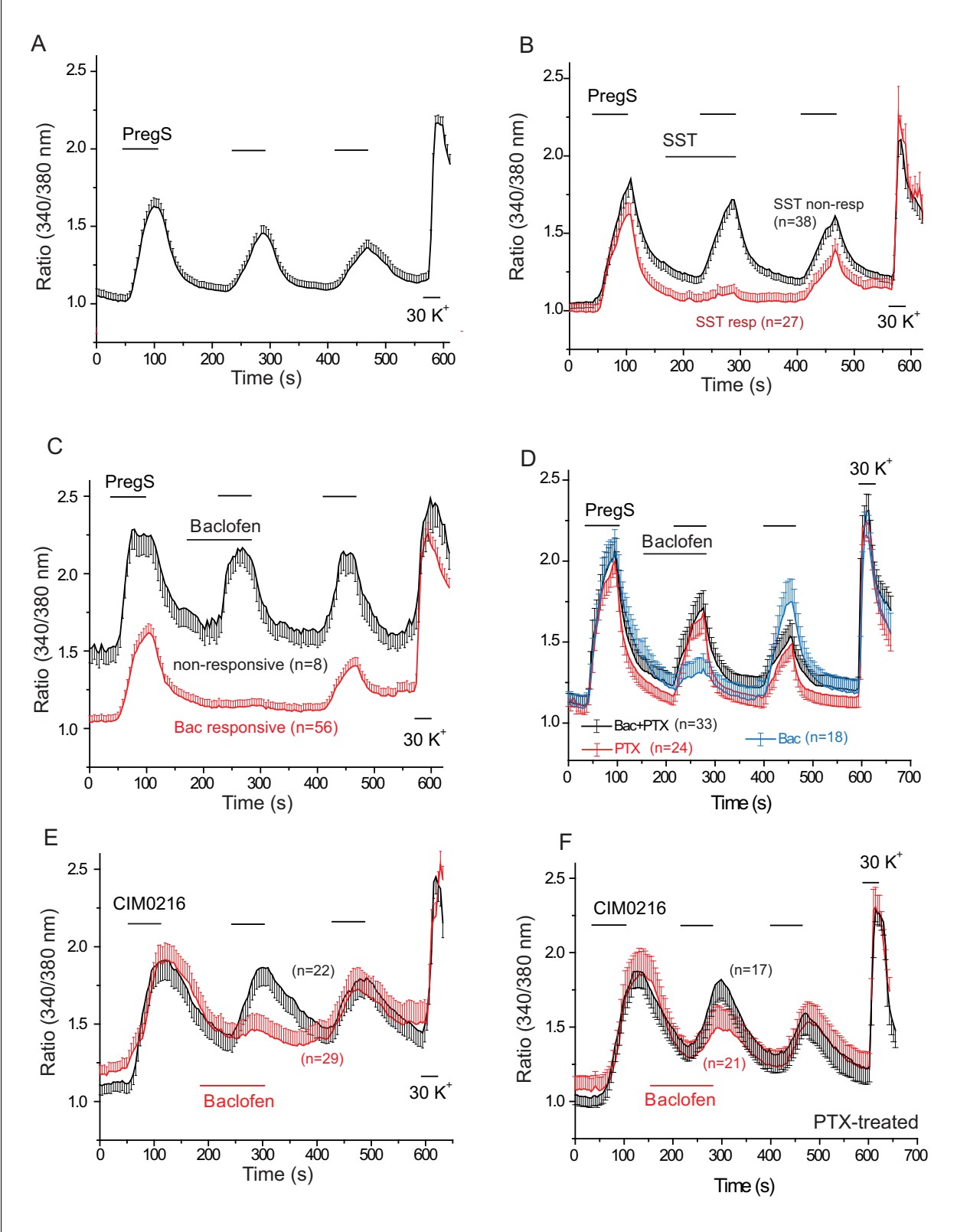

**Figure 4.** PregS-induced Ca$^{2+}$ signals are inhibited by agonists of Gi-coupled receptors in DRG neurons. Ca$^{2+}$ imaging experiments in DRG neurons were performed as described in Materials and methods. (**A**) Average trace ± SEM showing the effect of three consecutive applications of 12.5 μM PregS from neurons responsive to this compound; 30 mM KCl was applied at the end of the experiment. In (**B**) 1 μM somatostatin (SST) was applied before the second application of PregS, the two traces show the average ratios ± SEM in cells that responded to somatostatin (red) and in cells that did not

*Figure 4 continued on next page*

*Figure 4 continued*

(black). (**C**) Shows a similar measurement with 25 µM baclofen. (**D**) DRG neurons were treated overnight with 300 ng/ml PTX, the effects of 25 µM baclofen are compared in PTX treated (black) and non-treated (blue) cells. The red trace shows PTX treated cells without the application of baclofen. For these experiments, we pooled baclofen responsive and non-responsive cells, as cells not responding to baclofen would have been difficult to identify in the PTX treated group. (**E**) Measurements similar to panel C using the synthetic TRPM3 agonist CIM0216 (1 µM). Black trace is control cells not treated with baclofen, red trace represents baclofen treated cells. (**F**) Similar measurements to panel E in cells pretreated overnight with 300 ng/ml PTX; red trace represents baclofen treated cells, black shows control cells.

The following figure supplements are available for figure 4:

**Figure supplement 1.** Distribution of PregS responsive and non-responsive DRG neurons of TRPM8-GFP reporter mice.

**Figure supplement 2.** Individual traces and representative images for $Ca^{2+}$ imaging experiments.

**Figure supplement 3.** Baclofen does not inhibit PregS-induced $Ca^{2+}$ signals in non-neuronal cells, and $Ca^{2+}$ signals in DRG neurons evoked by KCl, the TRPM8 agonist WS12, or the TRPA1 agonist AITC.

Next, we tested the effect of the $GABA_B$ receptor agonist baclofen. *Figure 4C* shows that baclofen (25 µM) inhibited PregS-induced $Ca^{2+}$ signals in 87.5% of the neurons (56 out of 64). The effect of baclofen was strongly reduced by overnight pretreatment of the cells with pertussis toxin (PTX) (300 ng/ml), which ADP-ribosylates and thus inhibits $G\alpha i/o$ proteins (*Figure 4D*). The recently described more specific TRPM3 agonist CIM0216 (1 µM) also evoked clear $Ca^{2+}$ signals (*Figure 4E*) in many DRG neurons. Consistent with our data with PregS, baclofen also inhibited $Ca^{2+}$ signals evoked by CIM0216 in 87.8% of cells (29/33) (*Figure 4E*). In 4 cells, baclofen showed no inhibition of $Ca^{2+}$ signals evoked by CIM0216 (data not shown). Inhibition by baclofen was attenuated by pretreatment with PTX (*Figure 4F*). *Figure 4—figure supplement 2* shows representative images as well as representative traces for individual cells.

At the end of each experiment we applied 30 mM potassium chloride (KCl), to identify neurons. In *Figure 4* we only plotted data from neurons, defined as cells that responded to KCl with a robust $Ca^{2+}$ signal. A small number of KCl non-responsive, presumably non-neuronal cells, also responded to PregS, but baclofen did not inhibit PregS-induced $Ca^{2+}$ signals there (*Figure 4—figure supplement 3A*). In 42 individual experiments, 41 KCl negative cells responded to PregS (0–4 per cover slip); in the same experiments, 263 KCl-positive cells (neurons) responded to this TRPM3 agonist. In six experiments where CIM00216 was applied, 51 KCl positive cells (*Figure 4E*) and 6 KCl negative (not shown) responded to this compound. We did not investigate further this phenomenon and the exact nature of those PregS responsive non-neuronal cells, i.e. glia, or other cell types. We also found that baclofen had no effect on PregS-induced TRPM3 currents in Xenopus oocytes (data not shown), indicating that the drug did not directly act on TRPM3 channels.

TRPM3 is a non-selective cation channel, opening of which is expected to depolarize neurons and open voltage gated $Ca^{2+}$ channels (VGCC). Baclofen was shown to partially inhibit both high-, and low-voltage activated $Ca^{2+}$ channels in DRG neurons (*Huang et al., 2015*). To examine if this inhibition contributes to the effect of baclofen on PregS-induced $Ca^{2+}$ signals, we tested if this agent inhibits $Ca^{2+}$ signals evoked by 30 mM KCl. *Figure 4—figure supplement 3B* shows that baclofen did not induce any measurable inhibition of $Ca^{2+}$ signals evoked by KCl. Baclofen also did not inhibit $Ca^{2+}$ signals in DRG neurons evoked by the specific TRPM8 agonist WS12 (*Figure 4—figure supplement 3C*), which is consistent with earlier results showing that TRPM8 is not inhibited by the Gi-pathway (*Zhang et al., 2012*). Baclofen also did not inhibit $Ca^{2+}$ responses evoked by 25 µM allyl isothyocyanate (AITC, mustard oil), a TRPA1 channel agonist (*Figure 4—figure supplement 3D*). While higher concentrations of AITC (>100 µM), were reported to also activate TRPV1 (*Everaerts et al., 2011*), only 7 out of 62 AITC-responsive cells responded to the TRPV1 agonist capsaicin, and in the same experiments 35 cells responded to 0.5 µM capsaicin but not to AITC, which is consistent with AITC specifically activating TRPA1 at this concentration.

Functional $GABA_B$ receptors are obligate heterodimers of $GABA_{B1}$ and $GABA_{B2}$ receptors (*Padgett and Slesinger, 2010*). To test if the effect of baclofen depends on the presence of heterodimeric $GABA_B$ receptors, we co-expressed $GABA_{B1}$ and $GABA_{B2}$ receptors, with TRPM3 channels

in HEK293 cells (*Figure 5*). When both the $GABA_{B1}$ and $GABA_{B2}$ receptors were co-expressed with TRPM3, PregS-induced $Ca^{2+}$ signals were almost completely eliminated (*Figure 5A*). Upon washout of baclofen and PregS, a clear increase in $Ca^{2+}$ (off-response) was observed in most cells. The effect of baclofen was strongly alleviated by co-expression of the $G\beta\gamma$ sink $\beta$ARK-CT (*Figure 5A*), indicating the involvement of $G\beta\gamma$. Baclofen also essentially eliminated heat-induced $Ca^{2+}$ signals (*Figure 5B*); in these cells a marked off-response was also observed upon washout of baclofen. In cells expressing TRPM3 and only the $GABA_{B1}$ (*Figure 5C*) or only the $GABA_{B2}$ (*Figure 5D*) receptors,

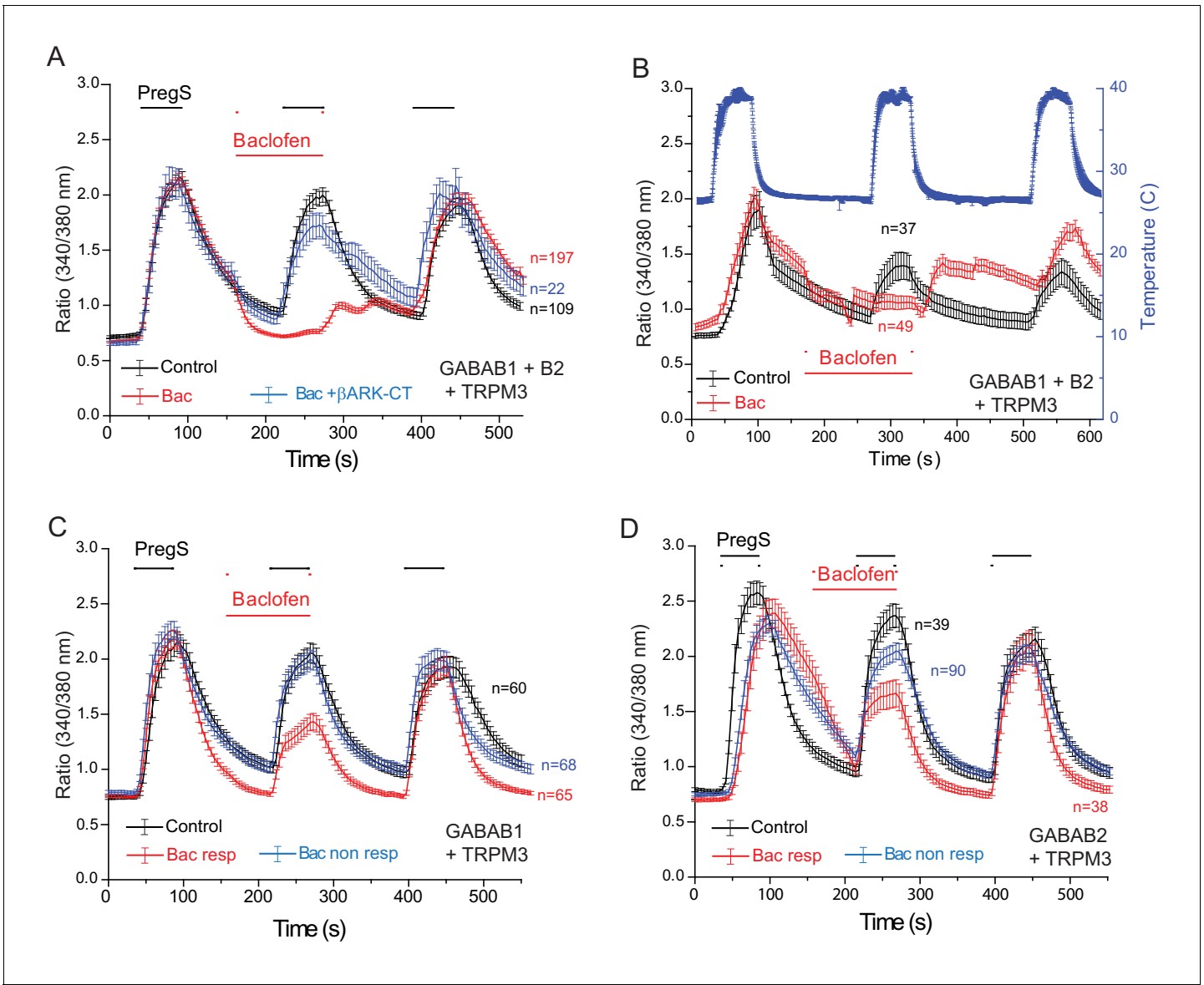

**Figure 5.** Baclofen inhibits PregS-induced $Ca^{2+}$ signals in HEK cells expressing the $GABA_{B1}$ and $GABA_{B2}$ receptors in a $G\beta\gamma$-dependent manner. $Ca^{2+}$ imaging experiments in HEK cells were performed as described in Materials and methods. Average traces ± SEM showing the effect of three consecutive applications of 12.5 µM PregS and the effect of 25 µM baclofen. The cells were transfected with mTRPM3 plus (**A, B**) $GABA_{B1}$ + $GABA_{B2}$ receptor, and in a subset of cells the $G\beta\gamma$ sink $\beta$ARK-CT (blue trace in panel A), (**C**) $GABA_{B1}$ receptor, (**D**) $GABA_{B2}$ receptor. In panel A, note the almost complete inhibition of PregS-induced $Ca^{2+}$ signal by baclofen, and the increase of $Ca^{2+}$ after washout of baclofen ('off' effect). In panel B, $Ca^{2+}$ responses to three consecutive heat pulses are shown (temperature: blue curve), note the marked off-response after washout of baclofen. In panels C and D the baclofen treated cells were subdivided into cells showing no response to baclofen (Bac non-resp), and cells in which baclofen induced a partial reduction of the PregS-induced $Ca^{2+}$ signals (Bac resp).

baclofen treatment only resulted in a small partial inhibition of PregS-induced $Ca^{2+}$ signals in a subset of cells.

Our data indicate that activation of three different endogenous Gi-coupled receptors inhibits native TRPM3 channels in DRG neurons. $Ca^{2+}$ signals, however, are not a linear readout of channel activity, thus we also performed whole-cell patch clamp experiments to confirm that activation of Gi-coupled receptors inhibit TRPM3 currents. To maximize our chances to obtain TRPM3 currents, we selectively patched small GFP positive neurons, most of which responded to PregS in $Ca^{2+}$ imaging experiments. Average capacitance in the control group was 7.55 pF, and in the baclofen-treated group, it was 8.63 pF; the majority of the selected cells (41 out of 43) responded to CIM0216. We focused on baclofen, as this agent induced inhibition in the highest proportion of neurons in our $Ca^{2+}$ imaging experiments. To avoid current desensitization, these experiments were performed in the absence of extracellular $Ca^{2+}$. *Figure 6* shows inward currents evoked by three repetitive applications of 5 μM CIM0216 in a nominally $Ca^{2+}$ free extracellular solution. In cells where baclofen was applied before the second CIM0216 pulse, the amplitude of the current was ~40% of the first pulse. Since current amplitudes also slightly decreased in control cells between the consecutive CIM0216 applications, this corresponds to a ~ 52% inhibition compared to the second CIM0216 application in control cells (*Figure 6B,C*). Inhibition of the CIM0216-induced currents by baclofen was reversible, as the third CIM0216 application evoked similar currents in control cells without baclofen treatment, and in baclofen treated cells after the drug was washed out. In the presence of 2 mM extracellular $Ca^{2+}$ inward currents induced by repetitive applications of CIM0216 showed a much more pronounced desensitization, decreasing to 35 ± 4% and 16 ± 5% of the first pulse in the second and third applications, respectively (n = 3).

## Baclofen inhibits nocifensive behavioral responses to the TRPM3 agonist CIM0216, but not responses to the TRPA1 agonist AITC

All our data so far was obtained on cell bodies of DRG neurons. GABA$_B$ receptors have been shown to be present not only at the central termini, but also at the peripheral processes of DRG neurons (*Hanack et al., 2015*). To assess if activation of GABA$_B$ receptors inhibits TRPM3 activity in the peripheral processes, we performed behavioral experiments. Injection of CIM0216 has been shown to induce nocifensive behavioral responses in mice (*Held et al., 2015*). We tested if these behavioral responses are inhibited by activation of GABA$_B$ receptors. We injected 50 nmoles/paw of CIM0216 into the hind paw of mice, and recorded nocifensive responses evoked by this compound. When baclofen (12.5 nmoles/paw) was coinjected with CIM0216, both the duration of licking, and the number of licks were significantly lower than in the group not injected with baclofen (*Figure 7A,B*). We also tested the effect of local baclofen injection on nocifensive responses evoked by hind paw injection of AITC. *Figure 7C,D* shows that baclofen did not significantly affect responses to this TRPA1 agonist.

## Discussion

Here, we provide evidence that TRPM3 channels are inhibited by activation of cell surface receptors that couple to Gi/o proteins via G$\beta\gamma$ subunits. The effect was robust, and showed no receptor specificity; activation of every recombinant and native Gi/o-coupled receptor we tested inhibited TRPM3 activity. Activation of heterologously expressed Gq-coupled receptors also inhibited TRPM3 via G$\beta\gamma$, but we focused on Gi-coupled receptors here to avoid confounding effects of concurrent PLC activation.

We found that in DRG neurons $Ca^{2+}$ signals evoked by TRPM3 agonists were inhibited in a subset of cells by activating Gi-coupled receptors with somatostatin, or the GABA$_B$ receptor agonist baclofen. The presence of non-responding cells for both agonists likely reflect cells not expressing the receptor, it is consistent with the high level of heterogeneity of DRG neurons, and also indicates that neither somatostatin nor baclofen is a direct inhibitor of TRPM3 channels. A much larger portion of DRG neurons responded to baclofen than to somatostatin, which correlates with the much higher expression level of GABA$_B$ receptors (*Thakur et al., 2014*). Baclofen also inhibited TRPM3 in a heterologous system co-expressing GABA$_{B1}$ and GABA$_{B2}$ receptors, in a G$\beta\gamma$-dependent manner. Baclofen also inhibited current responses to the TRPM3 agonist CIM0216 in DRG neurons, and in vivo nocifensive behavioral responses evoked by this TRPM3 agonist. G$\beta\gamma$ likely inhibits TRPM3 via direct

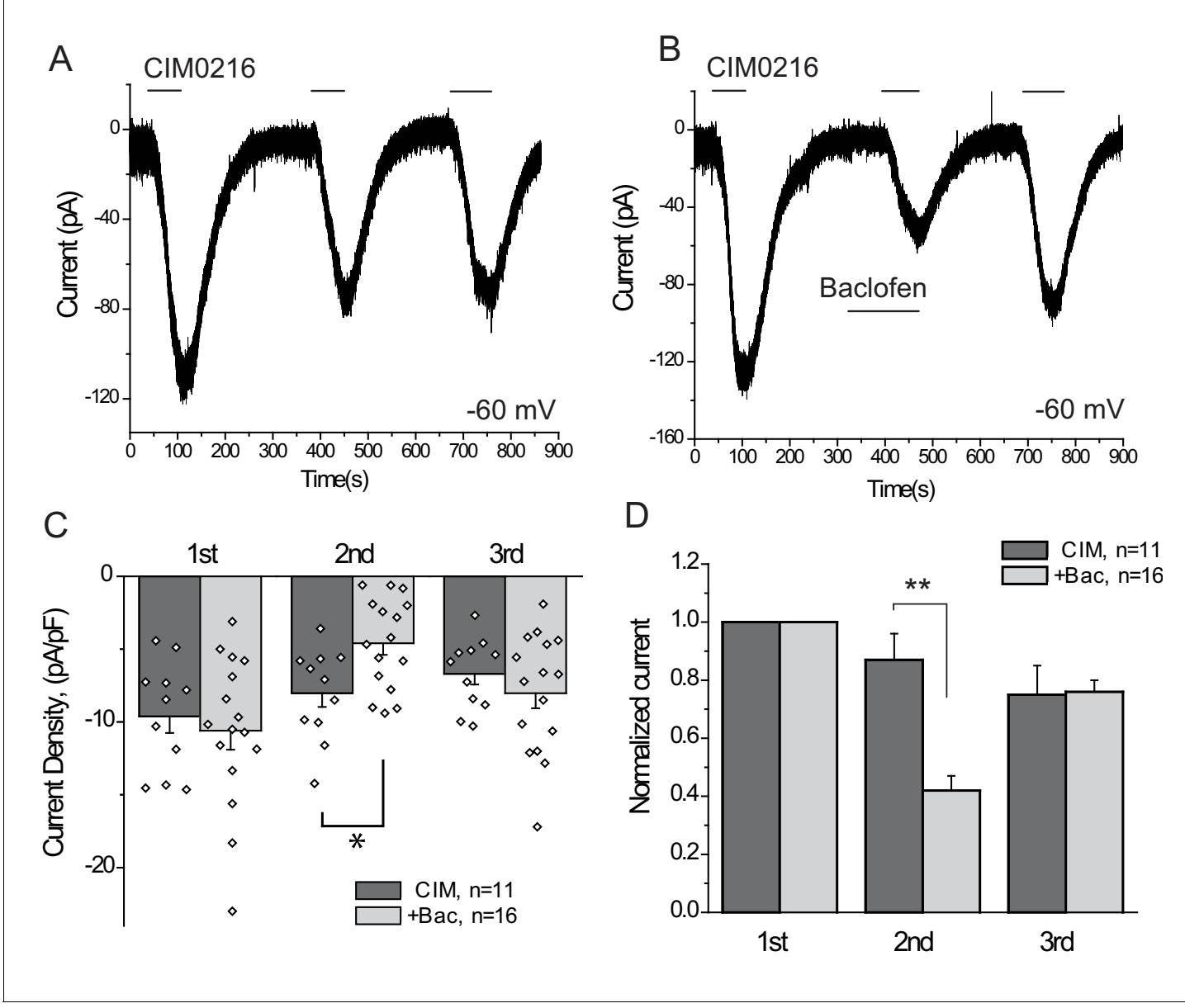

**Figure 6.** The GABA$_B$ receptor agonist baclofen inhibits inward currents induced by the TRPM3 channel agonist CIM0216. (A–B) Whole-cell patch clamp measurements in small GFP-positive DRG neurons were performed as described in Materials and methods at −60 mV holding potential in nominally Ca$^{2+}$ free solution. The applications of 5 µM CIM0216 and 25 µM baclofen are indicated by the horizontal lines. (C) Summary of current densities, (D) Summary of data normalized to the amplitude of the first peak current. Statistical analysis was performed with two sample t-test *p<0.05, **p<0.01.

interactions, because application of purified Gβγ protein to excised inside-out patches inhibited TRPM3, and we could detect biochemical interaction between the two proteins.

Gi-coupled receptors have two well-established ion channel targets, GIRK channels and N-type VGCC, both expressed in DRG neurons. Did the effect on those channels contribute to the effects of baclofen in behavioral experiments? While GIRK1 (KCNJ3) and GIRK2 (KCNJ6) channels expressed at relatively low levels in mouse DRG neurons (*Thakur et al., 2014*), we did not detect any outward currents in our patch clamp experiments in DRG neurons upon the application of baclofen. This may indicate that GIRK channels are not expressed at substantial levels in the same neurons as TRPM3,

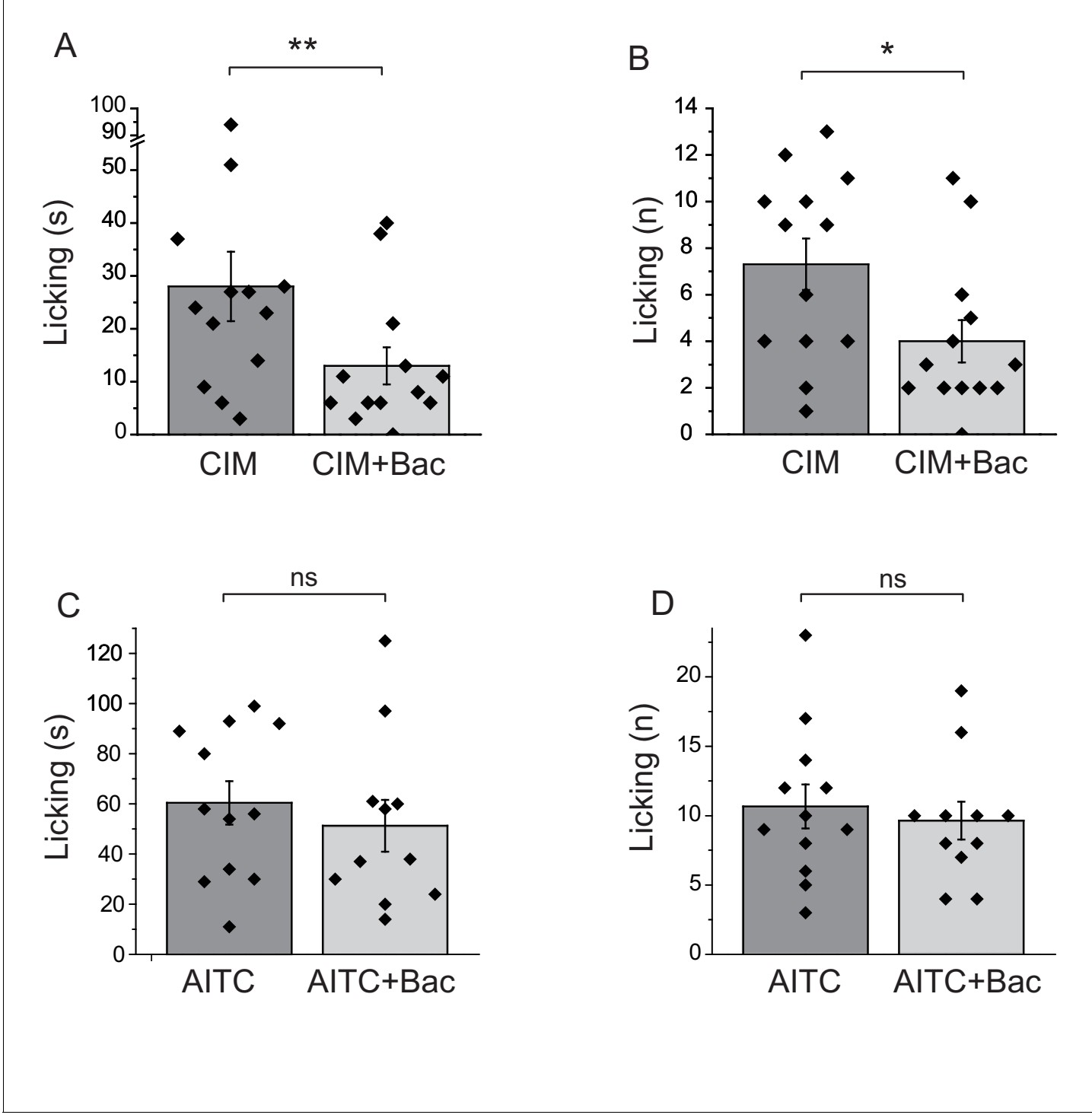

**Figure 7.** Baclofen inhibits nocifensive behavioral responses induced by the TRPM3 channel agonist CIM0216, but not responses to the TRPA1 agonist AITC. (**A–B**) Nocifensive responses to the injection of CIM0216 (50 nmol/paw) were recorded as described in Materials and methods in control animals, and in animals where 12.5 nmol/paw baclofen was also injected in the same hind paw. (**A**) Duration of licking, (**B**) number of licking (n = 13 for both groups). (**C, D**) Nocifensive responses to hind paw injection of 100 nmol/paw AITC were recorded as described in Materials and methods in control animals, and in animals where 12.5 nmol/paw baclofen was co-injected. (**C**) Duration of licking, (**D**) number of licking (n = 12 for AITC and n = 11 for AITC + baclofen). Statistical analysis was performed with two sample t-test *p<0.05, **p<0.01, ns: p=0.5 (**C**) and p=0.63 (**D**).

which is consistent with the finding that RNA for GIRK2 channels is enriched in the tyrosine hydroxylase expressing subpopulation of DRG neuron, which do not express TRPM3 (*Usoskin et al., 2015*).

Baclofen was also shown to inhibit both high- and low-voltage activated $Ca^{2+}$ channels in rat DRG neurons (*Huang et al., 2015*), but the effects were relatively modest, 32% and 22% inhibition, respectively. Interestingly, we did not detect any inhibition of high-potassium-induced $Ca^{2+}$ signals in DRG neurons by baclofen, in sharp contrast to the robust inhibition of $Ca^{2+}$ signals evoked by TRPM3 agonists. Among VGCCs, the N-type channels are classical targets of Gi-signaling; those channels are expressed in the central termini, and play role in transmitter release. We administered baclofen peripherally, thus it is unlikely that the behavioral effect of baclofen was due to inhibition of VGCC. We conclude that baclofen activates GABA$_B$ receptors in the peripheral processes and inhibits TRPM3 activity, and this inhibition is most likely responsible for the behavioral effect of baclofen.

Baclofen evoked a robust inhibition of $Ca^{2+}$ signals induced by the TRPM3 agonists PregS and CIM0216. In contrast, $Ca^{2+}$ signals evoked by the TRPM8 agonist WS12 (1 μM) and the TRPA1 agonist AITC (25 μM) were not inhibited by baclofen. While AITC was also shown to activate TRPV1 channels at higher concentrations (>100 μM), at 25 μM this compound does not activate TRPV1 (*Everaerts et al., 2011*). Nocifensive responses to hind paw injection of AITC were also not significantly affected by co-injection of baclofen. Similarly, activation of GABA$_B$ receptors by baclofen had no effect on $Ca^{2+}$ responses, inward currents and nocifensive responses evoked by the TRPV1 agonist capsaicin (*Hanack et al., 2015*). These data together show that GABA$_B$ receptor activation by baclofen, under basal conditions, specifically affects TRPM3 among thermosensitive ion channels in DRG neuron. Baclofen on the other hand was shown to inhibit inflammatory sensitization of TRPV1, as well as TRPV1-mediated thermal hyperalgesia during inflammation, in a non-G-protein-mediated manner (*Hanack et al., 2015*). Exploring the potential effect of baclofen on TRPM3 and other sensory ion channels in inflammatory conditions will require further research.

GIRK channels are activated by Gi/o-coupled receptors via direct binding of G$\beta\gamma$ subunits to the channel (*Logothetis et al., 1987*). Gq- or Gs-coupled receptors on the other hand do not activate GIRK channels in native cells or in expression systems (*Kobrinsky et al., 2000*), despite the general assumption that their activation also liberates G$\beta\gamma$. The mechanism of this selectivity between different G-protein pathways has been a subject for intensive research for more than two decades. The prevailing view by now is that GIRK channels form macromolecular complexes with Gi heterotrimers, and G$\beta\gamma$ rather than fully dissociating from G$\alpha$i, remains in the complex and activates the channel via a 'local conformational switch' and a surface masked by G$\alpha$i in the non-stimulated state, interacts with the channel (*Bünemann et al., 2003*; *Riven et al., 2006*). We find that TRPM3 inhibition does not show the G-protein isoform specificity characteristic of GIRK channels, as TRPM3 activity was inhibited by not only Gi-coupled receptors, but also by Gq-coupled receptors, at least in expression systems, and G$\beta\gamma$ sinks alleviated the inhibition by both groups of agonists. In this work, we focused on inhibition by the Gi/o pathway, and show that several endogenous Gi-coupled receptors in DRG neurons inhibit native TRPM3 currents. Exploring the effects of Gq-coupled receptor activation in native systems will require further studies.

An additional difference from GIRK channel activation is the following: GIRK channels when expressed in Xenopus oocytes display basal currents, which are due to free G$\beta\gamma$, and those basal GIRK currents are inhibited by co-expressing G$\alpha$i (*He et al., 1999*). In our hands PregS-induced TRPM3 currents were neither inhibited nor potentiated by the co-expression of G$\alpha$i3. GIRK channels are potentiated by G$\beta$1, $\beta$2, $\beta$3, and $\beta$4, but not by $\beta$5 subunits (*Mirshahi et al., 2002*); in our hands, TRPM3 was inhibited by G$\beta$1 but not by G$\beta$5. Overall, our data indicate that G$\beta\gamma$ inhibition of TRPM3 proceeds via a mechanism different from GIRK channel activation, but the two also share some common characteristics.

The closest relative of TRPM3 is TRPM1 (*Clapham, 2003*), which is expressed in retinal ON-bipolar cells, and its mutations in humans cause congenital stationary night blindness (*Irie and Furukawa, 2014*). In the dark, TRPM1 is kept closed by mGlur6 metabotropic glutamate receptors, which couple to heterotrimeric Go proteins. Upon light exposure decreasing glutamate levels lead to opening of TRPM1 (*Irie and Furukawa, 2014*). Both the G$\alpha$o and G$\beta\gamma$ subunits have been implied in inhibition of TRPM1, but their respective roles are controversial (*Koike et al., 2010a*, *2010b*; *Shen et al., 2012*; *Xu et al., 2016*). These controversies could be due to the fact that TRPM1 channels cannot be expressed reliably in heterologous systems, and native TRPM1 currents are small and difficult to differentiate from other endogenous channels (*Lambert et al., 2011*).

TRPM3 channels require $PI(4,5)P_2$ for activity, and inducible phosphatases that reduce the levels of this lipid inhibited TRPM3 activity, but this inhibition was partial and developed relatively slowly (*Badheka et al., 2015*; *Tóth et al., 2015*). We found that Gq-coupled receptor-mediated inhibition was not significantly alleviated by supplementing the whole-cell patch pipette with $PI(4,5)P_2$, even though activation of the receptor decreased $PI(4,5)P_2$ levels. The $G\beta\gamma$ 'sink' $\beta$ARK-CT on the other hand clearly attenuated the inhibitory effect of Gq-coupled receptor activation. While this result may sound puzzling, it indicates that upon GPCR activation $G\beta\gamma$ dominates over the reduction of $PI(4,5)P_2$ in inhibiting TRPM3 activity. Additionally, it is also possible that $PI(4)P$, which decreases much less upon GPCR-mediated PLC activation (*Borbiro et al., 2015*) may provide sufficient support to channel activity such that the additional $PI(4,5)P_2$ provided in the patch pipette will have no influence on channel activity. We found that activation of PDGFR, but not its PLC defective mutant, inhibited TRPM3 activity, indicating that, in principle, PLC activation alone may inhibit TRPM3 in conditions where $G\beta\gamma$ subunits are not released.

The $GABA_B$ receptor agonist baclofen inhibited TRPM3 activity in the vast majority of neurons we tested, and also inhibited behavioral nocifensive responses to a TRPM3 agonist. $GABA_B$ receptors are highly expressed in DRG neurons, and their activation has been shown to inhibit sensitization, but not basal activity of the heat and capsaicin sensitive TRPV1 channels in a non-G-protein mediated manner (*Hanack et al., 2015*). Various α-conotoxins such as Vc1.1, RgIA and PeIA were shown to inhibit N-type VGCC via a $GABA_B$ receptor activation in rat DRG neurons (*Adams et al., 2012*). Baclofen is often used as an adjuvant therapy in lower back pain; its effect is attributed to its central muscle relaxant properties (*Dapas et al., 1985*). The $GABA_B$ receptor agonists baclofen however has significant side effects such as drowsiness, mental confusion, muscle weakness (*Bowery, 2006*), and even paralysis and coma (*Caron et al., 2014*), which is not surprising, given the abundance of these receptors in the central nervous system (*Padgett and Slesinger, 2010*). Accumulating data showing that $GABA_B$ receptors inhibit activation or sensitization of nociceptive ion channels in DRG neurons raise the possibility of targeting this pathway for pain relief in the periphery.

## Materials and methods

### Whole-cell electrophysiology in HEK cells

Whole-cell patch clamp measurements were performed as described earlier (*Badheka et al., 2015*). Briefly Human Embryonic Kidney 293 (HEK293) cells were purchased from American Type Culture Collection (ATCC), Manassas, VA, (catalogue number CRL-1573), RRID:CVCL_0045; cell identity was verified by STR analysis. Passage number of the cells was monitored, and cells were used up to passage number 25–30, when a new batch of cells was thawed with low passage number; cells were tested for the lack of mycoplasma infection. The cells were transiently transfected with cDNA encoding the mouse TRPMα2 (mTRPM3α2) splice variant of Trpm3, in the bicistronic pCAGGS/IRES-GFP vector (*Oberwinkler et al., 2005*; *Vriens et al., 2011*), various GPCR constructs, and either the $\beta$ARK-CT (*Yamauchi et al., 2000*) or the Gαi3-G203A (*Ogier-Denis et al., 1996*) using the Effectene reagent (Qiagen). The cells were maintained in minimal essential medium (MEM) (Life Technologies, Carlsbad, CA, USA) supplemented with 10% (v/v) fetal bovine serum (FBS), 100 IU/ml penicillin and 100 µg/ml streptomycin. The cells were used for measurements 2 to 3 days after transfection at room temperature. Patch clamp pipettes were prepared from borosilicate glass capillaries (Sutter Instruments) using a P-97 pipette puller (Sutter Instrument) and had a resistance of 4–6 MΩ. Measurements were carried out on GFP positive cells, in an extracellular solution containing 137 mM NaCl, 5 mM KCl, 1 mM $MgCl_2$, 2 mM $CaCl_2$, 10 mM HEPES and 10 mM glucose, pH 7.4. The intracellular solution contained 140 mM potassium gluconate, 5 mM EGTA, 1 mM $MgCl_2$, 10 mM HEPES, and 2 mM Na-ATP, pH 7.3, adjusted with KOH. After a Giga-ohm seal was formed and the whole-cell configuration was established, the currents were recorded using a ramp protocol from −100 to +100 mV was applied once every second and the currents at −100 and +100 mV were plotted. The currents were measured with an Axopatch 200B amplifier, filtered at 2 kHz, digitized through Digidata 1322A and analyzed with pClamp 9.0 software (Molecular Devices).

## FRET-based monitoring of PI(4,5)P$_2$ hydrolysis

FRET measurements were performed as described earlier (*Borbiro et al., 2015*). Briefly, HEK cells were co-transfected with the CFP-tagged and the YFP-tagged tubby domain of the tubby protein, and either the human M1 or M2 muscarinic receptor. We used the R322H mutant of the tubby-based sensors, because this mutant is more sensitive to changes in PI(4,5)P$_2$ levels than the wild-type probes (*Quinn et al., 2008*). Fluorescence was detected using a photomultiplier-based dual-emission system mounted on an inverted Olympus IX-71 microscope. Excitation light (430 nm) was provided by a DeltaRAM light source (Photon Technology International, PTI). Emission was measured at 480 and 535 nm using two interference filters and a dichroic mirror to separate the two wavelengths. Data were analyzed with the Felix3.2 program (PTI). In *Figure 1—figure supplement 1* the ratio of the 535 and the 480 nm traces were plotted after normalizing to the ratio before the application of CCh.

## Ca$^{2+}$ imaging

Ca$^{2+}$ imaging measurements were performed with an Olympus IX-51 inverted microscope equipped with a DeltaRAM excitation light source (Photon Technology International, PTI), as described earlier (*Lukacs et al., 2013*). Briefly, DRG neurons or HEK cells were loaded with 1 µM fura-2 AM (Invitrogen) for 40 min before the measurement at 37°C, and dual-excitation images at 340 and 380 nm excitation wavelengths were detected at 510 nm with a Roper Cool-Snap digital CCD camera. Measurements were conducted in the same bath solution we used for whole-cell patch clamp, supplemented with 2 mM CaCl$_2$. PregS, baclofen, somatostatin and CIM0216 were applied with a gravity driven whole chamber perfusion system. Data analysis was performed using the Image Master software (PTI).

## Xenopus laevis oocyte preparation and RNA injection

Animal procedures were approved by the Institutional Animal Care and Use Committee at Rutgers New Jersey Medical School. *Xenopus laevis* oocytes were prepared as described earlier (*Rohacs, 2013*). Briefly, frogs were anesthetized in 0.25% ethyl 3-aminobenzoate methanesulfonate solution (MS222, Tricaine-S), (Western Chemical Inc, Ferndale, WA, USA) in H$_2$O (pH 7.4). Bags of ovaries were removed from the anesthetized frogs; individual oocytes were obtained by overnight digestion at 16°C in 0.1–0.2 mg/ml type 1A collagenase (Sigma-Aldrich), in a solution containing 82.5 mM NaCl, 2 mM KCl, 1 mM MgCl$_2$ and 5 mM HEPES (pH 7.4) (OR2). The next day the oocytes were washed multiple times with OR2 solution, then placed in OR2 solution supplemented with 1.8 mM CaCl$_2$ and 100 IU/ml penicillin and 100 µg/ml streptomycin and kept in a 16°C incubator. Linearized cRNA (30–35 ng) transcribed from the human TRPM3 (hTRPM3) cDNA clone (*Grimm et al., 2003*) in the pGEMSH vector and from G$\beta$1 and G$\gamma$2 (1 ng each) or various G$\alpha$i constructs (1 ng) were microinjected into individual oocytes. To have similar amount of RNA injected, RNA encoding GFP was co-injected with TRPM3 RNA in control oocytes. The injection was carried out with a nanoliter-injector system (Warner Instruments, Hamden, CT, USA). Oocytes were used for electrophysiological measurements 2–3 days after microinjection.

## Excised inside-out patch clamp and two-electrode voltage clamp (TEVC) electrophysiology

TEVC measurements were performed as described earlier (*Badheka et al., 2015*; *Lukacs et al., 2007*), briefly oocytes were placed in extracellular solution (97 mM NaCl, 2 mM KCl, 1 mM MgCl$_2$, 5 mM HEPES, pH 7.4) and currents were recorded with thin-wall inner-filament-containing glass pipettes (World Precision Instruments, Sarasota, FL, USA) filled with 3 M KCl in 1% agarose. Currents were measured with the same ramp protocol we used for excised inside-out patch measurements. The currents were recorded with a GeneClamp 500B amplifier and analyzed with the pClamp 9.0 software (Molecular Devices). To be able to compare data from experiments in different days, we normalized each day's data to the average PregS-induced current amplitudes in control TRPM3 expressing oocytes on the same day (*Figure 2D*). In each experimental day, one group was injected with G$\beta$1$\gamma$2 as a positive control, thus the larger number of experiments for that group, typically all experiments were performed on at least two different oocyte preparations and RNA injections.

Excised inside-out patch clamp measurements were performed as described earlier (*Badheka et al., 2015*; *Rohacs, 2013*). Briefly, oocytes were placed in bath solution (97 mM KCl, 5 mM EGTA, 10 mM HEPES, pH 7.4) in the recording chamber. The vitelline layer was removed with a pair of forceps, then giga-ohm seals were formed using borosilicate glass pipettes with resistance from 0.8 to 1 MΩ (World Precision Instruments, Sarasota, Florida, USA) containing pipette solution (97 mM NaCl, 2 mM KCl, 1 mM MgCl$_2$, 5 mM HEPES, 100 μM PregS, pH 7.4). Macroscopic currents were recorded with a −100 to +100 mV ramp protocol applied every second (0.25 mV/ms); holding potential was 0 mV. The currents were measured with an Axopatch 200B amplifier and analyzed with the pClamp 9.0 software (Molecular Devices, Sunnyvale, CA, USA). Test compounds, dissolved in bath solution, were applied to the cytoplasmic face of the membrane patch using a custom-made, gravity driven perfusion system. DiC$_8$ PI(4,5)P$_2$, was purchased from the Cayman Chemical Company (Ann Arbor, MI, USA). Purified Gβγ was purchased from two different sources. In the experiments shown in *Figure 3*, we used Gβγ purchased from Kerafast, recombinant mouse Gβ1 (ABK42205) and mouse Gγ2 (ABK42211.1) purified from SF9 cells, and recombinant rat Gαi1 (NP_037277.1) produced in High-Five Insect cells. Gαi1 was preactivated by incubating it with 100 nM GMP-PNP for 30 min on ice (*Koike et al., 2010b*). For *Figure 3—figure supplement 1* we used Gβγ, purified from Bovine Brain purchased from Merck Millipore. The stock solutions of this latter preparation contain 1250 ng of Gβγ in 25 μl buffer containing 0.1% lubrol, the final concentration of Gβγ in our experiments was 50 ng/ml, which resulted in a 0.0001% lubrol. Presumably due to the presence of this detergent, membrane patches were quite unstable in these experiments, and the seal was lost many times shortly after application of Gβγ.

## Immunoprecipitation and immunoblot

HEK293 cells on 6-well plates transfected with various constructs (indicated in *Figure 3E*) were harvested in lysis buffer (phosphate buffer saline with 5 mM EDTA and 0.5% Triton-X 100) supplemented with protease and phosphatase inhibitors. Myc-tagged-TRPM3 and Flag-tagged-Kir3.1 channels were immunoprecipitated by incubating pre-cleared cell lysates with primary anti-Myc (Cell Signaling, 2276S) or anti-Flag (Sigma, F3156) antibodies, respectively. The immune-complex was incubated with pre-washed protein G agarose beads overnight at 4°C with gentle-rocking. Immuno-precipitates were then used for Western blotting. After three washes, precipitates were eluted from the beads by incubating at 37°C for one hour in Biorad XT loading buffer and XT reducing agent. Protein samples were run on 4–12% Bis-Tris Criterion gels and transferred to PVDF membranes. The membranes were blocked at room temperature in TBS-T with 5% milk for 1 hr and then probed overnight at 4°C with a rabbit polyclonal anti-Gβ antibody (*Mirshahi et al., 2002*), recognizing Gβ1, Gβ2, Gβ3 Gβ4 (T-20, SC-378, Santa Cruz) diluted 1:500 in TBS-T with 5% milk. Secondary antibody used was donkey-anti-rabbit HRP (Thermo-Fisher, A16035) 1:5000 in 5% Milk. All blots were processed with SuperSignal West Pico Chemiluminescent Substrate (Thermo Fisher Scientific, Waltham, MA) and imaged with a Fuji Imager.

## DRG neuron isolation and culture

Animal procedures were approved by the Institutional Animal Care and Use Committee at Rutgers New Jersey Medical School. DRG neurons were isolated using the protocol based on Malin et al (*Malin et al., 2007*), with slight modifications as described previously (*Lukacs et al., 2013*; *Yudin et al., 2016*). Briefly, DRG neurons were isolated from adult mice of either sex (2–4 months old) from the TRPM8-GFP mouse line in C57/Blk background, expressing GFP driven by the promoter of TRPM8 (*Takashima et al., 2007*), see results for rationale. Mice were kept in a barrier facility under a 12/12 hr light dark cycle, with the light cycle starting at 7AM, a maximum of 4 mice were kept in the same cage, they were not subjected to any procedure, or drug administration before the experiments. Animals were anesthetized and perfused via the left ventricle with ice-cold Hank's buffered salt solution (HBSS; Invitrogen) followed by decapitation. DRGs were collected from all spinal segments after laminectomy and maintained in ice-cold HBSS during the isolation. After isolation and trimming of dorsal and ventral roots, ganglia were incubated in an HBSS-based enzyme solution containing 2 mg/ml type I collagenase (Worthington) and 5 mg/ml Dispase (Sigma) at 37°C for 25–30 min, followed by repetitive trituration for dissociation. After centrifugation at 80 × g for 10 min, cells were resuspended and plated on round coverslips pre-coated with poly-l-lysine (Invitrogen) and

laminin (Sigma), allowed to adhere for 1 hr and maintained in culture in in DMEM/F12 supplemented with 10% FBS for 12–36 hr before measurements (Thermo Scientific), 100 IU/ml penicillin and 100 µg/ml streptomycin and were kept in a tissue-culture incubator with 5% $CO_2$ at 37°C.

## Electrophysiology on DRG neurons

Whole-cell patch clamp recordings on DRG neurons were performed similar to that on HEK cells; we used small GFP-expressing neurons, and the external solution temperature was adjusted to 28–29°C. Neurons were perfused with a nominally $Ca^{2+}$ free bath solution containing 137 mM NaCl, 5 mM KCl, 1 mM $MgCl_2$, 10 mM HEPES and 10 mM glucose, pH 7.4 (adjusted with NaOH). Intracellular solutions contained 140 mM K-Gluconate, 1 mM $MgCl_2$, 2 mM $Na_2ATP$, 0.2 mM $Na_2GTP$, 5 mM EGTA, 10 mM HEPES, pH adjusted to 7.25 with KOH. In all experiments, cells that had a passive leak current more than 100 pA were discarded. Voltage-clamp recordings were performed at a holding potential of −60 mV and inward currents were evoked by 5 µM CIM0216.

## Behavioral test

One day before experiments, animals (2–4 months old) were transferred to the experimental room for acclimatization, separated in individual cages and provided food and water *ad libitum*. Experiments were performed during the day (light cycle). Stock solutions of CIM0216 (100 mM) alone or in mixture with Baclofen (25 mM) were diluted 20x in a sterile vehicle solution of 10% PEG-200 (Sigma-Aldrich), 2% Tween-80 (Amresco) in 0.9% NaCl. Hind paw subcutaneous dorsal injection of 10 µL was performed using a 30G needle coupled to a Hamilton syringe, and duration and numbers of nocifensive behavior (licking and lifting of injected paw) were recorded for a period of 10 min by an observer blind to the tested substances. Every animal was subject to random order injections of CIM0216 or its mixture with Baclofen in one paw followed 7 days later by the other reagent in opposite paw. A similar procedure was used for AITC and baclofen. We used both male and female animals and the data were pooled, as there was no significant difference between the two sexes. The final injected dose was 50 nmol/paw for CIM, 100 nmol/paw for AITC and 12.5 nmol/paw for Baclofen. All animal experiments were performed in accordance with the requirements of the Institutional Animal Care and Use Committee at Rutgers New Jersey Medical School

## Statistics

Data analysis was performed in Excel and Microcal Origin. Data collection was randomized. Behavioral experiments were blinded. No statistical method was used to predetermine sample sizes, but our sample sizes are similar to those generally employed by the field. The normality of the data was verified with the Kolmogorov-Smirnov test. Data were analyzed with t-test, or Analysis of variance *$p < 0.05$, **$p < 0.01$, ***$p < 0.005$. Data are plotted as mean +/- standard error of the mean (SEM) and scatter plots for most experiments.

## Acknowledgements

We thank Drs. Veit Flockerzi and Stephan Phillips for the mouse TRPM3α2 clone, Dr. Christian Harteneck for the human TRPM3 clone, Dr. Diomedes Logothetis for providing the various $G\beta\gamma$ sink clones. The GFP clone in the oocyte vector pGH-19 was a gift from Dr. Steven Siegelbaum (Columbia University).

## Additional information

### Funding

| Funder | Grant reference number | Author |
|---|---|---|
| National Institute of General Medical Sciences | R01GM111913 | Tooraj Mirshahi |
| National Institute of General Medical Sciences | R01 GM093290 | Tibor Rohacs |
| National Institute of Neurolo- | R01 NS055159 | Tibor Rohacs |

gical Disorders and Stroke

The funders had no role in study design, data collection and interpretation, or the decision to submit the work for publication.

## Author contributions

DB, Conceptualization, Formal analysis, Investigation, Visualization, Writing—review and editing, Performed most HEK cell and Xenopus oocyte electrophysiology experiments; YY, Conceptualization, Formal analysis, Investigation, Visualization, Writing—review and editing, Performed all $Ca^{2+}$ imaging, DRG neuron electrophysiology and behavioral experiments; IB, Formal analysis, Investigation, Visualization, Writing—review and editing, Performed the FRET and some of the HEK cell electrophysiology experiments; CMH, Investigation, Visualization, Designed and performed the coimmunoprecipitation experiments; AY, Formal analysis, Investigation, Performed some of the two-electrode voltage clamp experiments; TM, Conceptualization, Resources, Supervision, Funding acquisition, Writing—review and editing, Super-vised and designed the coimmunoprecipitation experiments; TR, Conceptualization, Formal analysis, Supervision, Funding acquisition, Visualization, Writing—original draft, Project administration, Writing—review and editing

## Author ORCIDs

Istvan Borbiro, http://orcid.org/0000-0002-8366-6881
Tibor Rohacs, http://orcid.org/0000-0003-3580-2575

## Ethics

Animal experimentation: Animal procedures were approved by the Institutional Animal Care and Use Committee (IACUC) at Rutgers New Jersey Medical School. Animals were handled according to the approved protocols #14056 (mice) and #14027 (frogs).

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
