## [Decision Letter]

Thank you for submitting your article "Inhibition of Transient Receptor Potential Melastatin 3 ion channels by G-protein βγ subunits" for consideration by *eLife*. Your article has been reviewed by three peer reviewers, and the evaluation has been overseen by Kenton Swartz as the Reviewing Editor and Richard Aldrich as the Senior Editor. The following individuals involved in review of your submission have agreed to reveal their identity: Thomas Voets (Reviewer #1); László Csanády (Reviewer #2); Alexander Chesler (Reviewer #3).

The reviewers have discussed the reviews with one another and the Reviewing Editor has drafted this decision to help you prepare a revised submission.

Summary:

This manuscript is one of three that reports exciting new findings on the mechanism of inhibition of TRPM3 channel activity in dorsal root ganglion (DRG) sensory neurons by stimulation of G protein-coupled receptors (GPCRs). Although all three manuscripts received favorable reviews, the essential revisions will require different amounts of time to address, and we encourage the authors to coordinate submission of their revised manuscripts.

This is an exciting study reporting that inhibition of TRPM3 ion channels by Gβγ subunits of heterotrimeric G proteins through a direct protein-protein interaction, and investigating the relevance of this inhibition to heat and pain sensation. The conclusion is that G-beta1-gamma2 dissociating from either Gα-i/o or Gα-q subunits is capable of inhibiting TRPM3 channels by a mechanism that is independent of membrane phosphatidyl-inositol-bisphosphate (PIP2) levels. The study employs a combination of co-immunoprecipitation (co-IP), electrophysiology, Ca^2+^ imaging, FRET, and behavioral studies in heterologous expression systems, dorsal root ganglion (DRG) neurons, and live animals. Strengths of the study are the logical buildup of hypotheses, the rigorous testing of each hypothesis by extensive experiments aimed at verifying possible alternatives, and the design of appropriate control experiments (e.g., ruling out involvement of PIP2 or Gα subunits in the studied phenomenon). Overall, this is an exciting and thorough study that is very appropriate for publication in *eLife*.

Essential revisions:

1) What is missing in this manuscript is an indication of the selectivity of the receptor activation towards TRPM3-dependent pain signals. To correlate the reduced pain response in the presence of baclofen to the inhibition of TRPM3 function, it is essential to show that other TRPM3-independent pain responses are not affected. This would involve demonstration that neuronal calcium signals and pain evoked by e.g. capsaicin or mechanical stimuli are not affected by baclofen.

2) To fully evaluate the quality and robustness of calcium imaging data it is helpful to have representative images as well as traces. This would allow the reader to appreciate the density and health of the cells and neurons as well as the robustness of the responses. Also, glia vs neurons in the culture should not be highlighted. Absence of KCl response is a poor proxy for glial identification – if this is the only criterion, then calling those cells KCl^-^negative cells is more appropriate.

3) In general, the scatter of data used to general bar graphs is unclear. These should be included on all summary plots throughout the manuscript to provide a better sense of the effect size and variability.

4) The description of the DRG imaging results in the text and Figure 4 is a bit confusing. It would be helpful to have these data plotted (e.g. as stacked bar or pie charts) so that the reader can more readily grasp what the total number of PS responsive sensory neurons is and what percentages are inhibited by the various GPCR agonists.

---

## [Author Response]

*Essential revisions:*

*1) What is missing in this manuscript is an indication of the selectivity of the receptor activation towards TRPM3-dependent pain signals. To correlate the reduced pain response in the presence of baclofen to the inhibition of TRPM3 function, it is essential to show that other TRPM3-independent pain responses are not affected. This would involve demonstration that neuronal calcium signals and pain evoked by e.g. capsaicin or mechanical stimuli are not affected by baclofen.*

The effects of GABA_B_ receptor activation by baclofen on TRPV1 channels in DRG neurons have been extensively characterized by Hanack et al., 2015. They demonstrated that in the basal, non-sensitized state, capsaicin-induced Ca^2+^ signals (Figure S3B) and inward currents (Figure 3) were not affected by baclofen treatment in DRG neurons, and behavioral responses in mice (FigS6H) were also not affected by local injection of baclofen. Figures refer to Hanack et al. We cite these aspects of that paper in detail now in the revised manuscript.

To test additional pain modalities, we also show in the revised version that amplitudes of Ca^2+^ signals evoked by the TRPA1 agonist allyl isothyocyanate (AITC, or mustard oil) (25 µM) were not reduced by baclofen in DRG neurons (Figure 4—figure supplement 3). Baclofen also did not have a significant effect on nocifensive responses evoked by hind paw injection of AITC (Figure 7).

We did not test mechanical responses, as it is difficult to evoke mechanically induced Ca^2+^ signals reliably in DRG neurons. Also, besides Piezo2, which is responsible for the rapidly adapting mechanically induced currents, the molecular identity of other mechanically activated currents, especially those responsible for mechanical nociception, is not yet established.

2) To fully evaluate the quality and robustness of calcium imaging data it is helpful to have representative images as well as traces. This would allow the reader to appreciate the density and health of the cells and neurons as well as the robustness of the responses.

We have included individual traces as well as representative images in Figure 4—figure supplement 2.

Also, glia vs neurons in the culture should not be highlighted. Absence of KCl response is a poor proxy for glial identification – if this is the only criterion, then calling those cells KCl^-^negative cells is more appropriate.

Thank you for the suggestion; these PregS-responsive cells correspond to a small fraction of KCl non-responsive cells, and thus may not necessarily be glial cells. We note this in the revised version, and call them KCl non-responsive, or non-neuronal cells.

*3) In general, the scatter of data used to general bar graphs is unclear. These should be included on all summary plots throughout the manuscript to provide a better sense of the effect size and variability.*

We have included scatter data in all figures where it was practical. For Figure 2 with the Two-electrode voltage clamp, we felt the figure would have been too crowded and confusing, we provide the scatter plot in Figure 2—figure supplement 1.

*4) The description of the DRG imaging results in the text and Figure 4 is a bit confusing. It would be helpful to have these data plotted (e.g. as stacked bar or pie charts) so that the reader can more readily grasp what the total number of PS responsive sensory neurons is and what percentages are inhibited by the various GPCR agonists.*

We have included a pie-chart in Figure 4—figure supplement 1, which shows the distribution of PregS-responsive cells.